**How much snow falls in the world's mountains?**
**A first look at mountain snowfall estimates in A-train observations and reanalyses.**
Anne Sophie Daloz[1,2,3], Marian Mateling[4], Tristan L'Ecuyer[2,4], Mark Kulie[5], Norm B. Wood[1],
Mikael Durand[6], Melissa Wrzesien[7], Camilla W. Stjern[3] and Ashok P. Dimri[8].
1.  Space Science and Engineering Center (SSEC), University of Wisconsin-Madison,

6        1225 West Dayton Street, 53706 Madison, WI, USA

2.  Center for Climatic Research (CCR), University of Wisconsin-Madison,

8        1225 West Dayton Street, 53706 Madison, WI, USA

3.  Center for International Climate Research (CICERO)

10       Gaustadalleen 21, 0349, Oslo, Norway

4.  Department of Atmospheric and Oceanic Sciences (AOS), University of Wisconsin-

12       Madison

13       1225 West Dayton Street, 53706 Madison, WI, USA

5.  NOAA/NESDIS/STAR/Advanced Satellite Products Branch

15       1225 West Dayton Street, Madison, WI 53706, USA

6.  School of Earth Sciences and Byrd Polar and Climate Research Center, Ohio State

17       University

18       108 Scott Hall, 1090 Carmack Rd, Columbus OH, 43210, USA

7.  Department of Geological Sciences, University of North Carolina at Chapel Hill

20       Chapel Hill, NC 25799, USA

8.  School of Environmental Sciences, Jawaharlal Nehru University,

22       New Delhi, 110067, India

**Abstract**

CloudSat estimates that 1773 cubic km of snow falls, on average, each year over the

world's mountains. This amounts to five percent of the global snowfall accumulations. This study
synthetizes mountain snowfall estimates over the four continents containing mountains (Eurasia,
North America, South America and Africa), comparing snowfall estimates from a new satellite
cloud radar based dataset to those from four widely used reanalyses: Modern-Era Retrospective
analysis for Research and Applications (MERRA), MERRA-2, Japanese 55-year Reanalysis (JRA-
55) and European Center for Medium-Range Weather Forecasts Re-Analysis (ERA-Interim).
Globally, the fraction of snow that falls in the world's mountains is very similar between all these
independent datasets (4-5%), providing confidence in this estimate. The fraction of snow that falls
in the mountains compared to the continent as a whole is also very similar between the different
datasets. However, the total of snow that falls globally and over each continent – the critical factor
governing freshwater availability in these regions – varies widely between datasets. The consensus
in fractions and the dissimilarities in magnitude could indicate that large-scale forcings may be
similar in the five datasets while local orographic enhancements at smaller scales may not be
captured. This may have significant implications for our ability to diagnose regional trends in
snowfall and its impacts on snowpack in the rapidly evolving alpine environments.






## 1. Introduction

Falling snow transfers moisture and latent energy between the atmosphere and the surface. Snow impacts the surface radiant energy transfer by modifying albedo and emissivity. Accumulated snow can also act as a thermal insulator that modifies sensible heat fluxes and how surface temperature responds to changes in atmospheric conditions. Furthermore, it acts as a surface water storage reservoir (Rodell et al., 2018), providing seasonal runoff that provides fresh water supplies for both human populations and water-dependent ecosystems. Billions of people around the world depends on these resources. These water supplies are recognized as being at risk from climate change and rising global temperatures (Barnett et al., 2005; Mankin et al., 2015).

The advent of satellite-borne instruments capable of detecting falling snow and of reanalysis products that diagnose snowfall have made possible a global examination of how snowfall is distributed and its contribution to atmospheric and surface processes. Precipitation gauge measurements of snowfall for meteorological and hydrological purposes provide valuable data but have historically suffered shortcomings related to spatial sampling and gauge performance (Kidd et al., 2017). Shortcomings in the accuracy of such measurements and methods to improve that accuracy have been the focus of a number of studies (Goodison et al., 1998; Kochendorfer et al., 2018). Beyond accuracy issues, these gauge networks are necessarily of limited spatial coverage potentially biasing climatologies over large domains. Coverage of ocean regions is not possible. Over land, gauges tend to be located near inhabited areas, leading to spare or nonexistent coverage in more remote locations (Groisman and Legates, 1994). These remote locations include areas such as the high latitudes and mountains, where snowfall can be the dominant form of precipitation. Even when these areas have relatively dense gauge networks such as the CONUS

(Contiguous United States) mountains, gridded datasets have their limitations, most notably gauge
under catchment issues and large snowfall accumulation gradients in complex terrain that are often
insufficiently sampled by existing in situ networks  (Henn et al., 2018).

Given these shortcomings in snowfall surface observations, studies on snowfall in remote
locations commonly rely on reanalyses (e.g. Bromwich et al., 2011). Reanalyses utilize numerical
weather prediction models to integrate observations of large-scale geophysical fields (e.g.,
temperature and water vapor). One strength of reanalysis datasets is their continuous spatial and
temporal coverage. However, the veracity of reanalysis snowfall datasets depends strongly on the
underlying model and the assimilated datasets, which often exhibits systematic and varied biases
(Daloz et al. 2018). In addition, their low spatial resolutions can be a limitation especially in
regions of complex topography and reanalyses should therefore be used with caution. For example,
Wrzesien et al. (2019) showed that reanalyses have large biases in terms of snow water equivalent
(SWE) over North America. Wang et al. (2019) compared the European Centre for Medium-Range
Weather Forecasts (ECMWF) Reanalysis 5$^{th}$ generation (ERA5) and ERA-Interim snowfall
estimates over Arctic sea ice and showed higher snowfall in ERA5 compared to ERA-Interim
resulting in a thicker snowpack for ERA5. Orsolini et al. (2019) focused on the Tibetan Plateau
and evaluated snow depth and snow cover estimates from reanalyses (ERA-5, ERA-Interim,
Japanese 55-year Reanalysis (JRA-55), and Modern-Era Retrospective analysis for Research and
Applications 2 (MERRA-2)), in situ observations and satellite remote sensing observations. They
showed that reanalyses can represent the snowpack of the Tibetan Plateau but tend to overestimate
snow depth or snow cover. Snow accumulation measurements from automatic weather stations are
compared to reanalysis datasets (ERA-Interim and National Center for Environmental Prediction
-2 (NCEP-2)) over the Ross Ice Shelf in Antarctica in Cohen and Dean (2013). While both
reanalysis datasets miss a number of accumulation events, ERA-Interim is able to capture more
events than NCEP-2. Liu and Magulis (2019) evaluated snowfall precipitation biases over Hign
Mountain Asia in MERRA-2 and ERA-5. The results show that, at high altitudes, snowfall is
underestimated in both reanalyses. In this current study, four reanalysis datasets will be examined:
MERRA, MERRA-2, ERA_Interim and JRA-55.

As an alternative to reanalyses, snowfall rates can now be assessed using satellite observations
(with sufficient spatio-temporal coverage) provided by CloudSat's Cloud Profiling Radar (CPR).
CloudSat observations, nearly continuous since 2006 (Stephens et al., 2002, 2008), have been
applied to produce near-global estimates of snowfall occurrence and intensity (Liu 2008; Kulie
and Bennartz, 2009; Wood and L'Ecuyer, 2018). The resulting datasets have been examined
extensively from local to global scales (Liu 2008; Kulie and Bennartz, 2009; Hiley et al., 2011;
Palerme et al., 2014; Smalley et al., 2015; Chen et al., 2016; Behrangi et al., 2016; Norin et al.,
2015; Milani et al., 2018; Lemonnier et al., 2019a, b). CloudSat has substantially extended the
spatial extent of precipitation measurements compared to existing gauge or radar networks. In
particular, these instruments have greatly enhanced the observations of light precipitation
including snowfall over oceans, over remote high latitude regions and over inaccessible land areas
(e.g., Behrangi et al., 2016; Milani et al., 2018; Smalley et al., 2015; Norin et al., 2017; Lemonnier
et al. 2019a, b).

However, satellite-based retrievals also have inherent uncertainties related, for example, to
their limited temporal coverage. For instance, they might miss some heavy events such as
atmospheric rivers in Western North and South America (Ralph et al., 2005; Neiman et al., 2008;
Viale and Nunez, 2011). Therefore, CloudSat snowfall retrievals have been extensively assessed
against a wide range of independent ground-based measurements. Hiley et al. (2011) seasonally
compared CloudSat snowfall estimates with Canadian surface gauge measurements, showing
better results for higher versus lower latitudes - especially lower latitude coastal sites. They
speculated that the latitudinal differences might be due to CloudSat sampling (more observations
at higher latitudes), snow microphysical differences associated with warmer snow events that
could affect CloudSat estimates (e.g., wetter snow, rimed snow, and/or mixed phase precipitation),
or precipitation phase identification issues associated with snow events in the 0-4°C temperature
range. CloudSat's 2C-SNOW-PROFILE (2CSP) product also displayed excellent light snowfall
detection capabilities when compared against the National Multi-Sensor Mosaic QPE System
(NMQ) dataset, a hydrometeorological platform which assimilates different observational
network. Still, CloudSat did not produce higher snowfall rates as frequently as NMQ (Cao et al.,
2014). Further comparisons between CloudSat and the National Centers for Environmental
Prediction (NCEP) merged NEXRAD and rain gauge Stage IV dataset illustrated consistent
CloudSat-Stage IV performance when near-surface temperatures are below freezing (Smalley et
al., 2014). The CloudSat 2CSP product was also compared to a ground-based radar network in
Sweden, showing consistent agreement in the $0.1 - 1.0$ mm h$^{-1}$ snowfall rate range (Norin et al.,
2015). However, 2CSP snowfall rate counts were lower above the 1 mm h$^{-1}$ threshold. 2CSP
retrievals have also been rigorously compared to ground-based profiling radars in Antarctica, with
CloudSat outperforming ERA-Interim grid-averaged results when MRR-derived retrievals are
used as a reference dataset (Souverijns et al., 2018). Comparisons between CloudSat and existing
reanalysis datasets are however scarce, and mostly limited to the poles (Palerme et al., 2014, 2017;
Milani et al., 2018; Behrangi et al., 2016). Together, these independent analyses provide
confidence that CloudSat observations may deliver realistic accumulations on seasonal scales. The
CloudSat snowfall dataset has also been proven useful for isolating distinct modes of snowfall
variability on global scales. For instance, over-ocean convective snow has been comprehensively
studied using CloudSat products (Kulie et al., 2016; Kulie and Milani, 2018). CloudSat also
exhibits enhanced snowfall observational capabilities in mountainous regions compared to ground-
based radar networks, partially due to scanning radar beam blockage issues (Smalley et al., 2014).

In spite of the noted shortcomings in snowfall datasets from gauge, radar and reanalyses,
mountain snowfall has not yet been thoroughly studied using multiple reanalyses and the CloudSat
data set. In this study, we derive mountain snowfall from five datasets (CloudSat 2CSP, MERRA,
MERRA-2, ERA-Interim and JRA-55) to answer the following questions:
1. How much snow falls on the World's mountains?
2. What percentage of continental snow falls on mountainous regions?
Given the challenges in retrieving snowfall from single-frequency radar observations, especially
in complex terrain, the CloudSat estimates are not treated as the "reference" dataset, though we
note that they are the only estimates derived directly from observations. All five sources are treated
as providing valid independent estimates of the fraction of snow that falls in mountainous
compared to all continental regions to document the current state of knowledge in this field. The
next section presents the different datasets employed in this study, as well as methodological
information such as the mountain and continental masks. Section 3 compares mountain snowfall
fraction and magnitudes between the different datasets while the following section, Section 4
discusses the differences in absolute magnitude of snowfall estimates. Finally, Section 5
summarizes the results of this study and offers concluding remarks.

**2. Data and Methodology**
**2.1 Satellite observations**
For this work, the CloudSat data are spatially gridded onto a $1^o$ x $3^o$ (lat/lon) grid. The
nadir-pointing CPR onboard NASA's CloudSat satellite is the first spaceborne W-band (94-GHz)
radar. CloudSat's high inclination orbit ($98^o$) provides a unique coverage of observed global
snowfall (Kulie et al., 2016). In addition to providing near-global sampling, the CPR has a
minimum detectable radar reflectivity of approximately -29 dBZ and is consequently sensitive to
lighter precipitation events (Tanelli et al., 2008). The CPR has a fixed field of view pointed at
near-nadir and measures over a spatial resolution of approximately 1.7 km along-track and 1.4 km
cross-track (Tanelli et al., 2008). The orbit is such that CloudSat revisits particular locations every
16 days. While this observing strategy limits sampling on short time-scales, CloudSat has observed
more than 120 million snowing profiles over its 10+ year mission providing a rich dataset from
which to derive snowfall frequency and cumulative snowfall over the large domains analyzed here.
CloudSat data are available from 2007.

CloudSat's 2CSP snowfall product, version R04 (Wood et al., 2013), provides estimates
of instantaneous surface snowfall rates (S) for each of these pixels derived from the observed
vertical profiles of radar reflectivity (Z). A version R05 is now available however, the snow
retrieval status variable is evaluated in the same way in the two versions of the product. The global
snowfall amount is very similar in R04 and R05 so the results should only differ slighly with the
new version of CloudSat. The data are spatially gridded onto a 1° x 3° (lat/lon) grid to ensure robust
sampling by the narrow CloudSat ground track. This means that the satellite data are sampled onto
the spatial grid desired and then averaged within each grid. The product derives instantaneous data
twice per month from an optimal estimation retrieval (Rodgers, 2000). They are then applied to
individual reflectivity profiles to obtain vertical profiles of snow microphysical properties. Ground
clutter affects radar bins nearest the surface, so the retrieval is applied only to the clutter-free
portion of the profile, i.e., that portion of the profile that is above the extent of likely ground clutter
effects, typically about 1.2 km over land. Surface snowfall rate is estimated as the rate in the lowest
clutter-free radar bin. The cumulative snowfall presented here are, thus, not true surface snowfall
rates. Grazioli et al. (2017) compared the vertical profile of precipitation from the ECMWF
Integrated Forecasting System (IFS) model with satellite-borne radar measurements. They showed
some noticeable differences between the different datasets in the vertical structure. Clutter limits
CloudSat's ability to detect shallow snow events or capture strong variations in snow profiles near
the surface (Maahn et al, 2014; Souverijns et al, 2018; Palerme et al, 2017). While this introduces
uncertainty in the snowfall estimates presented here, the analysis of ground-based vertically-
pointing radar in East Antarctica and in Svalbard (Norway) by Maahn et al. (2014) show that the
effects of this observing system limitations may be compensated by the competing effects of
evaporation and undetected shallow snowfall. It should also be noted that on November 1 2011,
there was a change in CloudSat's operating mode, leading to daytime-only operations, which can
lead to some uncertainty in the snowfall estimates.

Snow and rain are discriminated based on the CloudSat 2C-PRECIP-COLUMN product
(Haynes et al., 2013), which applies a melting layer model driven by the ECMWF analyses
temperature profiles. Snow particles are assumed to melt following the model of melted mass
fraction described by Haynes et al. (2009). All profiles with melted fractions less than about 15%
at the surface (<1.2km) are considered snowing. Those with melted fractions greater than 90% are
considered raining. Melted/frozen fractions between 15-90% are labeled "mixed" category
considered to be a catch-all uncertainty for profiles that cannot be unambiguously classified as rain
or snow using W-band reflectivity alone. Only snowing profiles are considered in this study.

**2.2 Reanalyses**

This study also considers four modern reanalyses: MERRA, MERRA-2, ERA-Interim and

JRA-55. MERRA (Rienecker et al., 2011; 0.67° x 0.5° x 42 levels) uses the Goddard Earth
Observing System version 5 (GEOS-5) and the data assimilation system (DAS). MERRA-2
(Gelaro et al., 2017; Bosilovich et al., 2015; 0.635° x 0.5° x 42 levels) was recently introduced to
replace MERRA. ERA-Interim (Dee et al., 2011; 0.75° x 0.75° x 37 levels) is developed by the
ECMWF. ERA-Interim replaced the previous reanalysis dataset from the ECMWF, ERA-40. The
Japanese Meteorological Agency (JMA) has recently developed their second reanalysis dataset
after JRA-25: JRA-55 (Kobayashi et al., 2015; 0.56° x 0.56° x 60 levels). Both MERRA
(Rienecker et al., 2011) and MERRA-2 (Gelaro et al., 2017) use 3D variational assimilation
systems, where JRA-55 (Kobayashi et al., 2015) and ERA-Interim (Dee et al., 2011) use 4D. The
spatial and temporal modeling of snowfall alone is different in these reanalyses, as are some of the
physical mechanisms within. The MERRA-2 reanalysis is based on an updated version of the
GEOS-5 atmospheric model. Reichle et al. (2017) showed that the snow amounts are generally
better represented in MERRA-2 than MERRA. However, MERRA-2 precipitation has a known
deficiency over high topography due to issues in categorizing precipitation mode as large-scale
instead of convective (Gelaro et al., 2017). The results from these previous studies make the
comparison between MERRA and MERRA-2 particularly interesting in this case. JRA-55
assimilates the same observations that were used for the predecessor to ERA-Interim, ERA-40, as
well as archived observations from JMA. Both JRA-55 and ERA-Interim use their own forecast
models.

CloudSat has not been assimilated in any of the four reanalyses so it can be considered as
independent. All datasets used in this study are bilinearly interpolated from their native resolution
to match the $1^o$ x 3° (lat x lon) grid of CloudSat. The data are examined over the time period 2007-
2016 with a monthly temporal resolution. The production of MERRA data ended in February 2016,
as MERRA-2 is now the preferred dataset while CloudSat started in 2007.

**2.3 Masks and definitions**
Snowfall estimates from all sources are partitioned between the different continents using
the "continental mask" shown in Figure 1a. The continental mask was first used in L'Ecuyer et al.
(2015). Then, the mountain and non-mountain regions are separated using the "mountain mask"
presented in Figure 1b. Based on the Kapos et al. (2000) definition, grid cells are classified as
mountainous based on elevation, slope, and local elevation range. They used the global digital
elevation model GTOPO30 and ARC-INFO to identify areas above particular altitudes and
generate grids containing the slope and the local elevation range. Then, they combined these
variables, with adapted criteria, to define mountainous regions. The original mask was produced
using with a spatial resolution of 30 arc-seconds (~1 km). Our version of the mountain mask has
been aggregated to 1° x 3° (lat/lon) grid to match the spatial resolution of the gridded CloudSat
2SCP. The combination of these two masks is used to subdivide the snowfall estimates over the
four continents that contain mountains: North America, South America, Eurasia and Africa.

In this article, total mountain snowfall is equal to the cumulative snow falling over North
America, South America, Africa and Eurasia. Greenland and Antarctica are considered as ice
sheets and therefore do not qualify as continents with mountains. Global snowfall is the cumulative
snow falling over all lands in the world, which includes the four continents already cited plus
Greenland, Australia and Antarctica.

**3.  Mountain snowfall estimates in CloudSat observations and reanalyses**
**3.1 Global spatial distribution of mountain snowfall**
Table 1 shows the snowfall estimates for mountain and non-mountain snowfall for
CloudSat and the reanalyses, over each continent and globally. According to CloudSat
observations, 1773 cubic km of snow falls over global mountains per year. This number is an
average over the volume of snow falling during the time period from 2007 to 2016. From CloudSat
estimates, 5% of global snowfall is within mountainous areas. It is encouraging that the fraction
of snow falling in the mountains occupies a narrow range from 4% for MERRA's reanalyses and
JRA-55 to 5 % for ERA-Interim and CloudSat. This good agreement between the different datasets
(Table 1) allows us to state with some confidence that 5% of all continental snow falls in the
mountains globally. In the reanalyses, while the fraction of snow within the mountains is similar
across all datasets, the amount of snow falling over the mountains varies depending on the dataset
examined (cf. Table 1). MERRA and MERRA-2 global mountain snowfall estimates are close to
CloudSat with 1763 and 1891 cubic km per year, respectively, while ERA-Interim and JRA-55
show much lower amounts, with 1041 and 489 cubic km per year, respectively.

To visualize where the snow is falling, Figure 2 presents the geographical distribution of
the mountain snowfall estimates in CloudSat and the reanalyses. As expected, in all datasets a
majority of the mountain snow falls in the Northern Hemisphere (Himalayas and Rockies; 95-
99%), with little snowfall (<5%) in the Southern Hemisphere. The geographical patterns exhibited
by MERRA, MERRA-2 and CloudSat seem to resemble each other while ERA-Interim and JRA-
55 tend to show different geographical distributions with generally lower snow rates. However,
when focusing on specific regions, we can see that MERRA-2 has also major differences compared
to MERRA and CloudSat: For example, over South America or Eastern Russia, MERRA-2
produces much more snow than all the other datasets. Another interesting difference appears when
comparing the datasets over North America versus Asia. ERA-Interim has higher snow rates in
the Rockies compared to the Himalayas while for the other datasets they are comparable. To go
deeper into the comparison of the datasets, Figure 3 presents the differences in geographical
distribution of mountain snowfall between CloudSat and the reanalyses over the High-mountain
Asia. This figure clearly shows very large differences between CloudSat and the reanalyses,
reaching +/- 10 mm/month/gridbox at some locations. In general, both ERA-Interim and JRA-55
present much lower snow accumulations compared to CloudSat. On the other hand, MERRA and
MERRA-2 present lower snow accumulations on the southern part of the domain and higher on
the northern part. These differences in snowfall distribution have major implications in terms of
mountain runoff, millions of people in the surrounding regions depend on these resources. The
systematically lower mountain snowfall estimates in ERA-Interim and in JRA-55, as well as the
tendency for MERRA-2 to produce higher mountain snowfall rates over some continents will be
further discussed below.

**3.2 Contribution of mountain snowfall to continental snowfall**
Table 1 also shows the contribution of mountain snowfall to total snowfall for CloudSat
and each reanalysis over each continent. To get a better sense of the contribution of orography to
snowfall, the percentage of mountainous grid points over each continent is provided in the last
column of the table. Eurasia has the highest fraction of mountainous grid boxes with 33% of its
grid boxes considered as mountains. North and South America have a quarter of their grid boxes
covered with mountains and only 14% of the African continent is considered mountainous. The
contribution of mountain snowfall does not vary substantially between continents. For Eurasia,
South America and Africa, it is around 10 % while for North America it represents around 5% of
the snow falling over the continent. Over all the continents, the agreement between the reanalyses
and CloudSat observations is very good with differences under 4%.

Coherently with the previous section, the magnitude of mountain snowfall estimates over
the four continents vary a lot depending on the datasets examined. MERRA's datasets and
CloudSat present similar magnitude in terms of mountain and continental snowfall while ERA-
Interim and JRA-55 present much lower estimates than the other datasets. For example, over
Eurasia the values for mountain snowfall vary between 379 for JRA-55 and 1440 cubic km per
year for CloudSat. Over North America, it varies from 105 cubic km per year for JRA-55 to 378
cubic km per year for MERRA-2 and for South America from 5 for JRA-55 to 86 cubic km per
year for MERRA-2. Unfortunately, the high range of differences observed for mountain snowfall
also applies for the magnitude of total snowfall over each continent. In all cases, JRA-55 shows
the lowest magnitude estimates and MERRA-2 the highest. It is also interesting to point out that
CloudSat is always part of the higher range of snowfall estimates for each continent. Due to its
limited temporal coverage, it might be missing some heavy snow events such as atmospheric rivers
in Western North America (Rutz and Steenburgh, 2012; Lavers and Villarini, 2015; Molotch et al.
2010). These few events contribute to a large part of the water year precipitation but as the analysis
has been done over several years, this should have a limited impact on the total accumulated snow.

**4. Examination of the differences in snowfall magnitude**
The previous section showed a very good agreement between all the datasets in terms of
mountain snowfall fractions. However, the spatial maps presented in Figure 2 and the absolute
snowfall amounts in Table 1 showed substantial differences in magnitude between the different
datasets. This is further demonstrated in Figure 4 that summarizes the snowfall estimates in
mm/month/grid box over Eurasia, North America, South America and Africa and its partitioning
between mountainous (blue) and non-mountainous areas (yellow) for the five datasets. To ease the
comparison between the different datasets, here the snowfall amounts are normalized by the
number of mountain and non-mountain grid boxes respectively. There is some consistency in the
relative behavior of the various datasets between the regions. Consistently with the results in
Section 3, JRA-55 always has the lowest estimates of snowfall per grid box (cf. Table 1). For
example, over North America and Eurasia, JRA-55 produces 68% less snowfall than the average
of the four other datasets (Fig. 4). Even so, when looking at Figure 5, which presents the frequency
of snowfall occurrences for each continent for all datasets, the frequency of snowfall occurrences
for JRA-55 is very close to the other products. This indicates that JRA-55 underestimates the
intensity of many snowfall events. ERA-Interim also tends to be on the lower end of the spectrum
concerning snowfall, compared to the other datasets (Fig. 4). This can be at least partly attributed
to its systematic lower frequency of snowfall occurrences (cf. Figure 5). With the exception of
North America, MERRA-2 generally has the highest total snowfall compared to the other datasets
(Fig. 4). Again, this is consistent with the results shown in the previous section. This overestimate
is related to the way this dataset represents the frequency of snowfall events. MERRA-2 produces
much more snowfall events than the other datasets (cf. Figure 5). This bias might be similar to the
bias identified for precipitation in climate models, producing too frequent and too lightly-
precipitating events, referred to as "perpetual drizzle" (Stephens et al., 2010). This could be
happening for snowfall events in MERRA-2.

The differences in snowfall among datasets is especially prominent over Africa and South
America. Over Africa (Fig. 4d), both MERRA and MERRA-2 produce much more snow than the
other datasets, with MERRA-2 producing nearly twice as much snowfall as MERRA. MERRA
produces 75% more snowfall than the average of the three remaining datasets (ERA-Interim, JRA-
55 and CloudSat) while MERRA-2 produces 85% more. For the same reasons, over South America
MERRA-2 produces 73% more snowfall than the average of the other datasets. Furthermore, it
highly exceeds the mountain and non-mountain snowfall compared to the other datasets. However,
as most of the snow over South America is mountainous, the excess in mountainous snowfall has
a stronger impact on the differences in total accumulated snowfall. The seasonal cycle of mountain
snowfall over South America (not shown) provides another interesting explanation for this specific
bias. From January to December, MERRA-2 overestimates the other datasets but with a similar
seasonal cycle in the first part of the year. However, during the second part (after June), the
behavior of MERRA-2 is very different – instead of a decrease in mountain snowfall, snowfall
accumulations remain very high and steady. This is clearly a major contributor to the high snowfall
estimates of MERRA-2 over South America.

Overall, these results are coherent with previous studies comparing different reanalysis

datasets (Daloz et al. 2018, Sebastian et al. 2016, Thorne and Vose 2010). They all show that
reanalyses are able to represent some general patterns but also show very important differences.
For example, Sebastian et al. (2016) compared atmospheric budgets for the computation of water
availability in different reanalyses. They showed considerable variations in the individual
components of the different budgets and suggested that part of these variations could be attributed
to differences in the representation of clouds and convective schemes for precipitation.
Furthermore, Daloz et al. (2018) showed significant differences in the representation of clouds in
the reanalyses examined in this article, confirming the hypothesis of Sebastian et al. (2016). More
specifically, they showed that JRA-55 exhibits some strong deficiencies in the representation of
clouds and that MERRA-2 introduces some biases that were not evident in MERRA. These results
may partly explain the deficiencies observed for these two datasets.

**5. Summary and conclusion**

Snowfall plays an important role in a number of atmospheric and surface processes that

impact energy and hydrological cycles and can influence Earth's climate. To understand these
processes, and how they will be influenced by future climate change, it is imperative to have
reliable observations of present-day mountain snowfall. This study is a preliminary step towards
an estimate of mountain snowfall from CloudSat satellite observations and four reanalyses
(MERRA, MERRA-2, JRA-55 and ERA-Interim). In this work we answer the following questions:

1.  How much snow falls on the World's mountains?

1773 cubic km per year of snow falls on the World's mountains in CloudSat observations, 1763
cubic km per year in MERRA, 1891 cubic km per year in MERRA-2, 1041 cubic km per year in
ERA-Interim and 489 cubic km per year in JRA-55 (cf. Table 1).

2. What percentage of continental snow falls on mountainous regions?

4 to 5% of snow falls over the mountains (cf. Table 1).

One aim of this research is to provide context for researchers who want to use snowfall

estimates globally or on specific continents from reanalyses and/or satellite observations. The
results of the discussion clearly emphasize the necessity of using several datasets, including
different platforms such as reanalyses and satellite observations. Results presented here can help
future analyses select validation datasets for specific continents, since we show that some datasets
behave differently than the others for continental snowfall estimates. For instance, modelers have
difficulties accurately representing snowfall over South American mountains (Gelaro et al., 2017),
and it is suspected that MERRA-2 is not the optimal dataset to use for this continent. However,
this study and Wrzesien et al. (2019) showed that over North America, MERRA-2 is certainly a
realistic dataset with substantial skills. Generally, there is no good or bad dataset, however some
datasets may outperform others over certain continents. These different abilities in the reanalyses
and satellite products can lead to issues when validating climate models, for example. We therefore
recommend to use an ensemble of the products just like it is recommended to use several models
or simulations. This study also suggests that estimates of the fraction of snow that falls in the
mountains compared to all-continental snowfall may be more reliable than estimates of the
absolute magnitude of mountain snow accumulations. A hypothesis behind this result could be that
the datasets presented here have a similar representation of the large-scale forcings but differences
at local/smaller scales, which could be due to differences in the physical parameterizations of the
models, subgrid-scale parameterizations of orographical effects. Indeed, even if the reanalyses are
based on different models, they should simulate similar and realistic large-scale forcings. For
CloudSat, its ability to capture these forcings would come from its relatively good level of temporal
and spatial coverages. This could explain the consensus between the different datasets in terms of
snowfall fractions. On the other hand, at smaller scales, both types of datasets experience different
limitations which would explain the dissimilarities in snowfall magnitude. For example, for
CloudSat, its spatial coverage could lead it to miss some heavy snow events like atmospheric
rivers.

In the future, this work will expand in several directions. First, a deeper and more process-

oriented analysis of the differences observed during the different datasets should be done over each
continent. While this study is confined to mountain snowfall produced by CloudSat and reanalysis
datasets, it also serves as a foundation for studying cloud microphysical and dynamical processes
operating within snow-producing clouds forced by orography. Because different modes of
snowfall have varying impacts on the environment and potentially unique remote sensing
fingerprints, identifying specific types of snowfall could lead to better measurements of snowfall.
In addition, this could also improve forecasting by representing different snowfall modes more
realistically within numerical weather models. Also, to evaluate the ability of climate models to
represent snowfall estimates, this same analysis could be realized for climate models such as the
CMIP5 ensemble, or the forthcoming CMIP6 ensemble.

**Acknowledgments and Data**
For ASD, this research was supported by a seed grant from the Center of Climatic Research of the
University of Wisconsin-Madison. Parts of this work by TL was performed at the University of
Wisconsin-Madison for the Jet Propulsion Laboratory, California Institute of Technology,
sponsored by the National Aeronautics and Space Administration (NASA) CloudSat program.
This work by MK was also partly supported by a NASA grant award NNX16AE21G. Parts of this
research by NBW were performed at the University of Wisconsin - Madison for the Jet Propulsion
Laboratory, California Institute of Technology, sponsored by the National Aeronautics and Space
Administration. CloudSat data used herein were acquired from the CloudSat Data Processing
Center (DPC) and at the time of writing can be accessed online at
http://www.cloudsat.cira.colostate.edu; we acknowledge the support of the DPC in providing the
data. MERRA and MERRA-2 data were provided by NASA's Global Modeling and Assimilation
Office (GMAO) and obtained through the Goddard Earth Sciences Data and Information Services
Center (GES DISC). JRA-55 data was provided by the Japanese Meteorological Agency and
obtained through the National Center for Atmospheric Research's (NCAR) Research Data
Archive. ERA-Interim data was provided by the European Centre for Medium-Range Weather
Forecasts (ECMWF) and obtained via the ECMWF WebAPI. The views, opinions, and findings
contained in this report are those of the author(s) and should not be construed as an official
National Oceanic and Atmospheric Administration or U.S. Government position, policy, or
decision.

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

**Tables**

| Snowfall estimates | MERRA | MERRA-2 | ERA-Interim | JRA-55 | CloudSat | Percentage of mountain grid boxes per continent |
|---|---|---|---|---|---|---|
| **Eurasia** | 1416 /11176 **11%** | 1426 / 13104 **10%** | 808 /8112 **9%** | 379 / 3916 **9%** | 1440 / 10764 **12%** | 33% |
| **North America** | 312 / 4500 **6%** | 378/5800 **6%** | 223 /3450 **6%** | 105 / 1725 **6%** | 303 / 7325 **4%** | 24% |
| **South America** | 30 / 270 **10%** | 86 / 662 **12%** | 10 / 100 **9%** | 5 / 46 **10%** | 30 / 236 **11%** | 21% |
| **Africa** | 0.5 / 6 **8%** | 0.8 / 11 **7%** | 0.1 / 1 **9%** | 0.07 / 0.5 **12%** | 0.2 / 2 **9%** | 14% |
| **Global** | 1763/ 43403 **4%** | 1891/47127 **4%** | 1041/21363 **5%** | 489/11288 **4%** | 1773/35027 **5%** | |

**Table 1:** The table summarizes the snowfall estimates of mountain and non-mountain snowfall for MERRA, MERRA-2, ERA-Interim, JRA-55 and CloudSat for the time period 2007-2016, for Eurasia, North America, South America, Africa and globally. Global snowfall is the cumulative snow falling over all lands in the world, which includes the four continents already cited plus Greenland, Australia and Antarctica. For each area and dataset, a table cell shows: the amount of mountain (top left), non-mountain snow (top right; cubic km per year) and the contribution of mountain snow to the total amount of snow falling over a continent (bottom, %). The last column shows the percentage of grid boxes considered as mountain by the mountain mask over each continent.

**Figures**

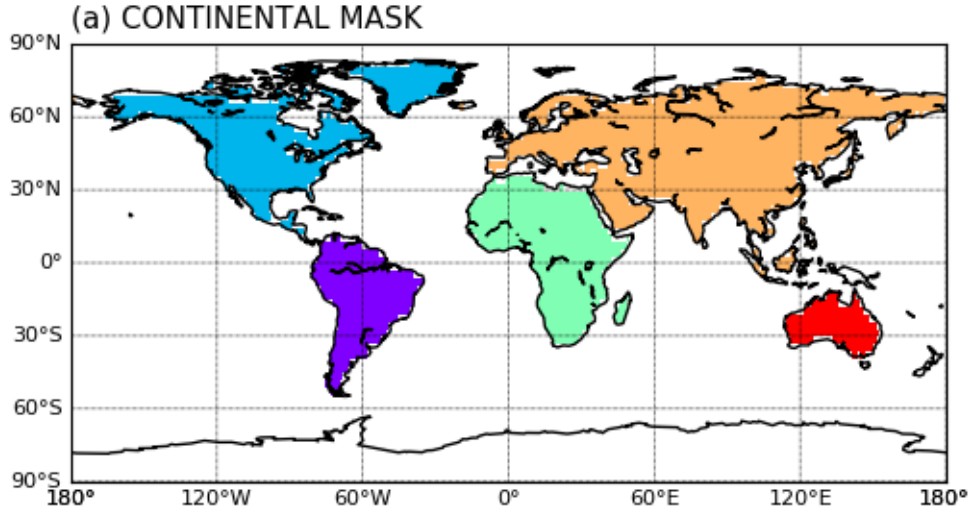

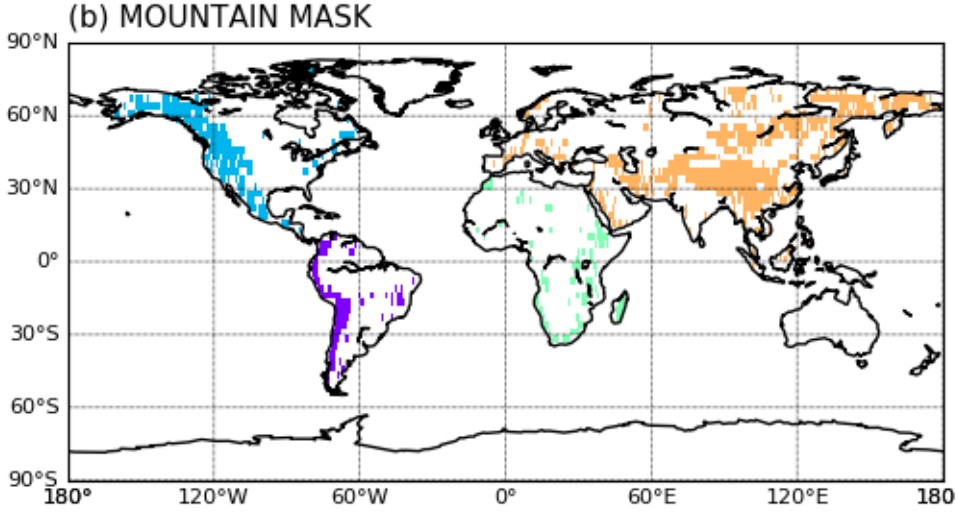

**Figure 1:** Spatial maps of the continental mask (a) with specific colors for each continent: blue for
North America, pink for South America, orange for Eurasia, green for Africa, red for Australia
and white for Antarctica; and the associated mountain mask (b) for each continent containing
mountains.

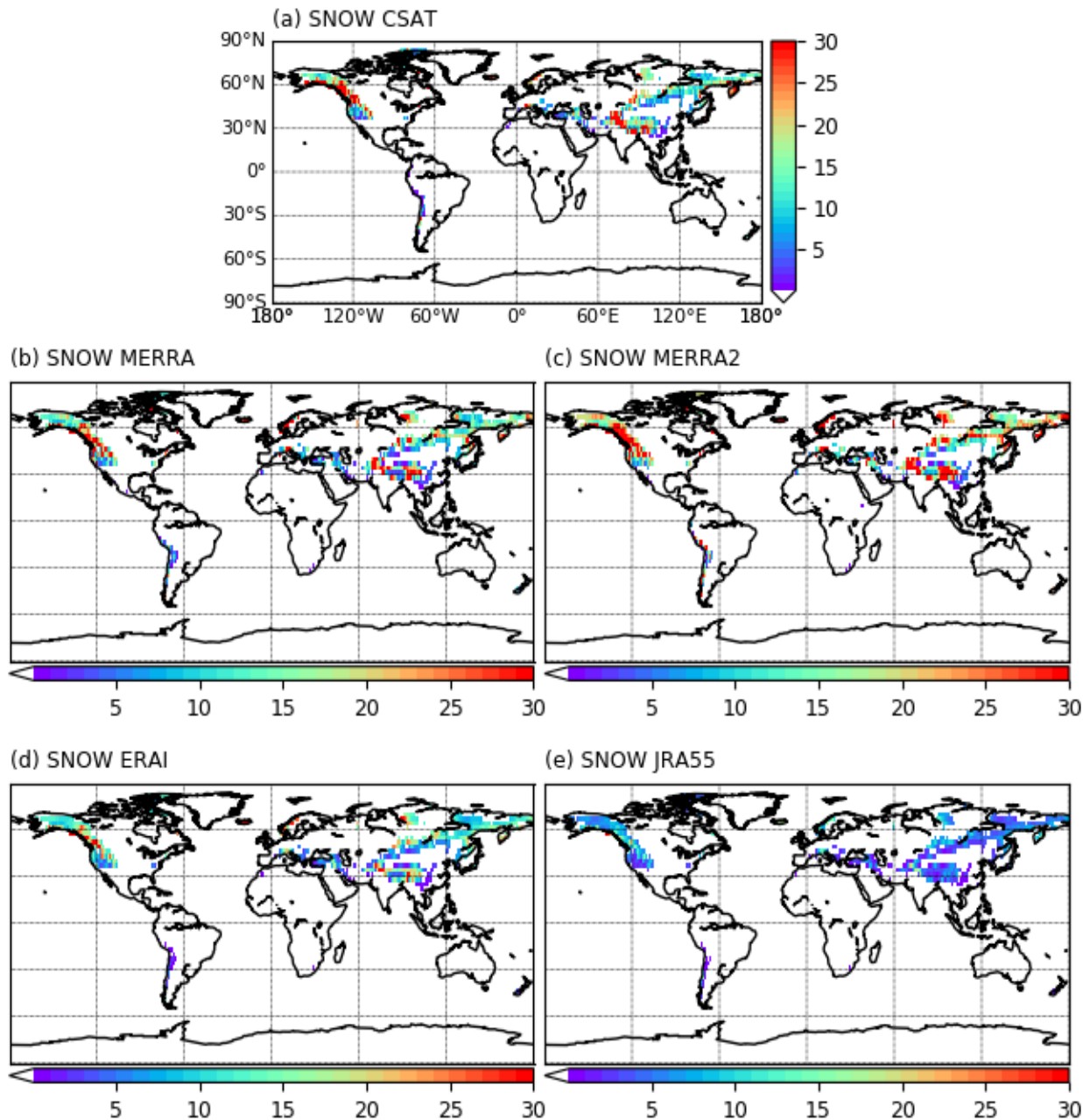


**Figure 2:** Spatial maps of global cumulative mountain snowfall (mm/month/gridbox) for a) CloudSat, b) MERRA, c) MERRA-2, d) ERA-Interim and d) JRA-55, averaged over the time period 2007-2016.



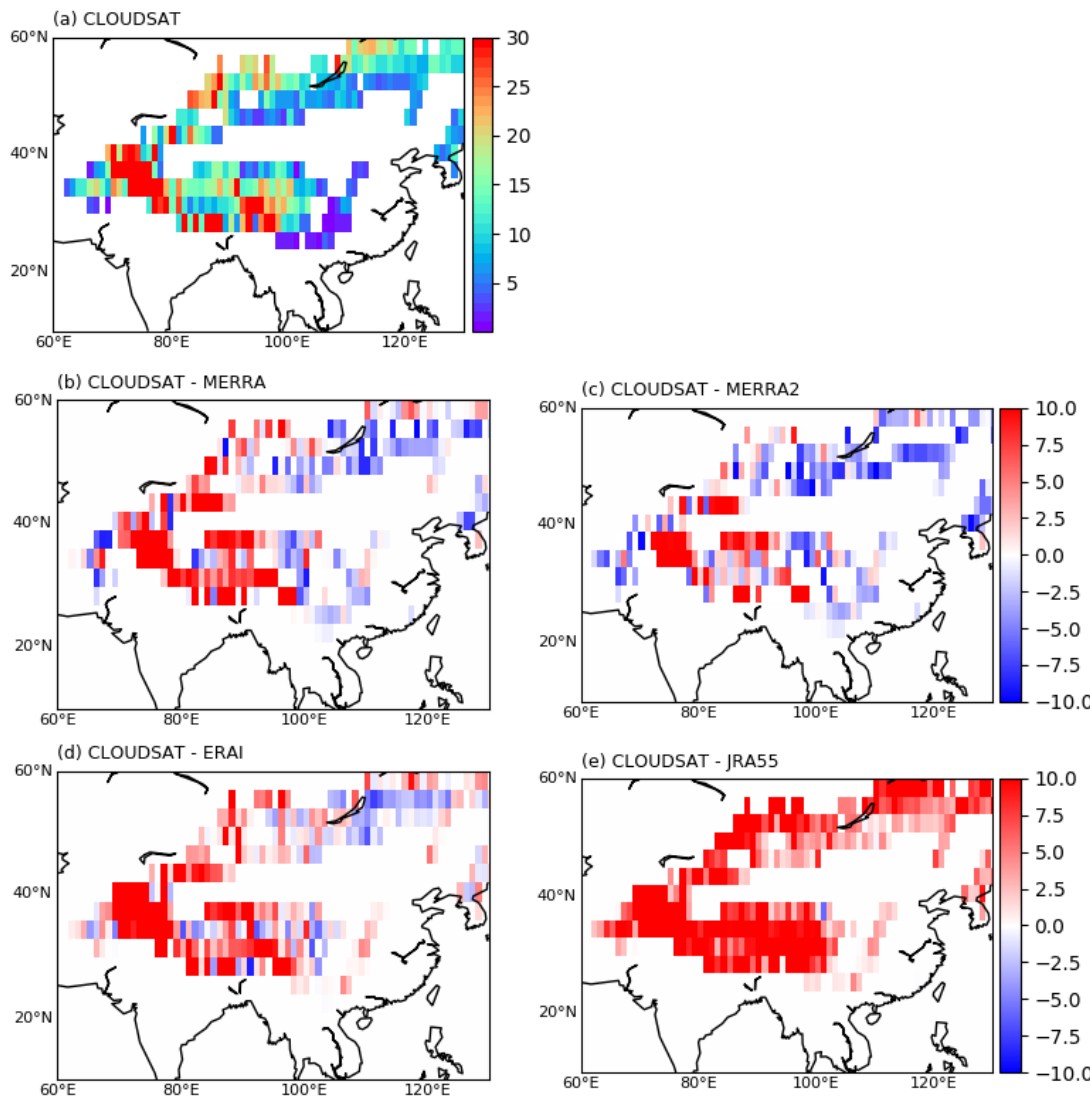


**Figure 3:** Spatial maps of the global cumulative mountain snowfall (mm/month/gridbox) over the

High-mountains Asia for a) CloudSat, b) CloudSat minus MERRA, c) CloudSat minus MERRA-

2, d) CloudSat minus ERA-Interim and d) CloudSat minus JRA-55, over the time period 2007-

2016.




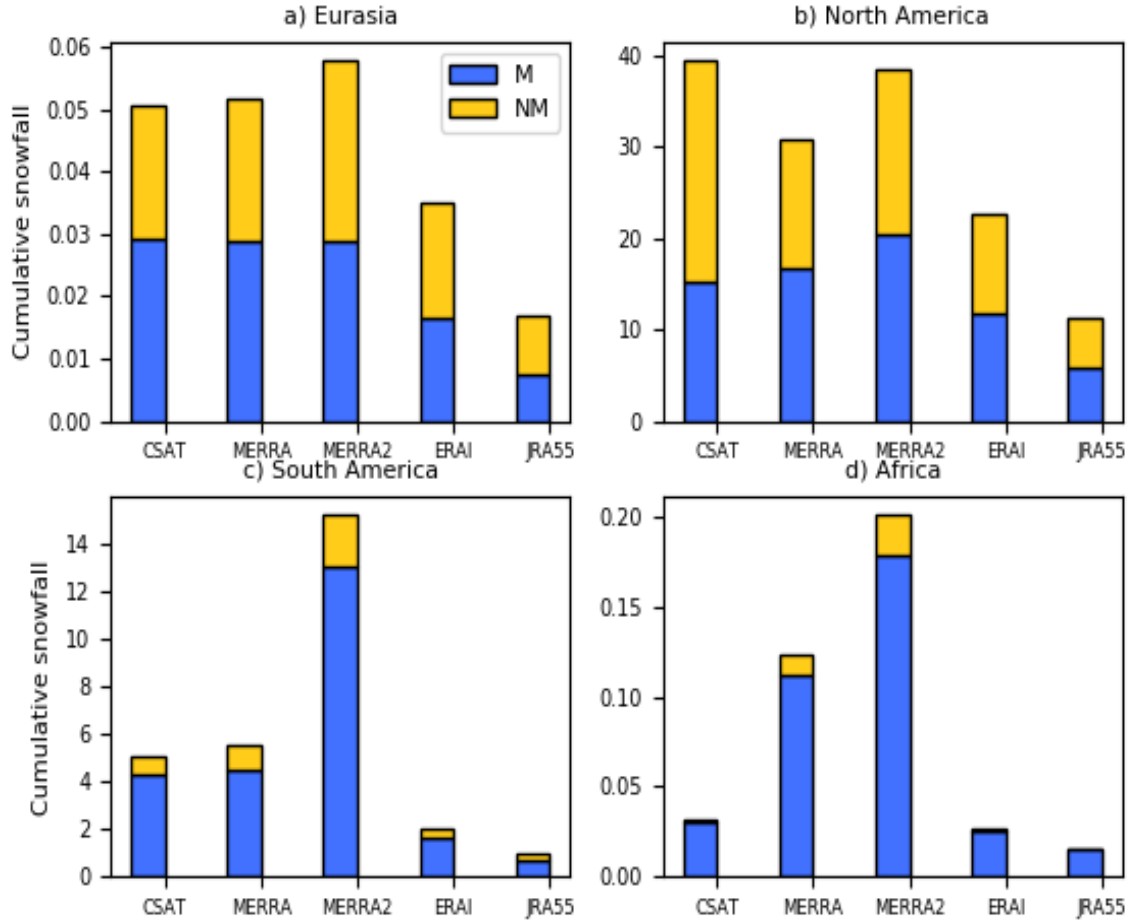


**Figure 4:** Snowfall estimates (mm/month/grid box) over: a) Eurasia, b) North America, c) South

America and d) Africa for CloudSat, MERRA, MERRA-2, ERA-Interim and JRA-55 over the

time period 2007-2016. Mountain snow is in blue and non-mountain snow is in yellow.


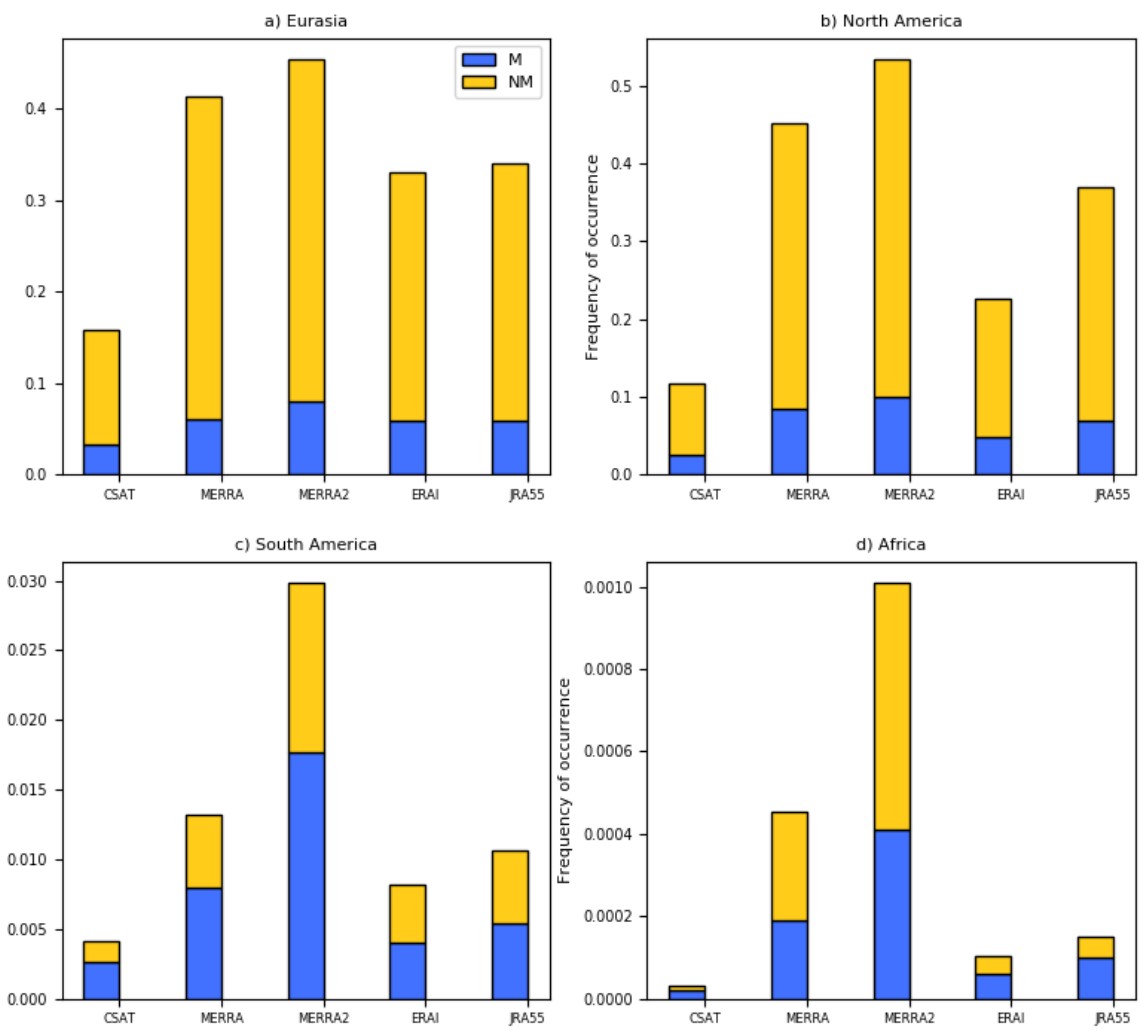


**Figure 5:** Frequency of occurrence of snowfall estimates over: a) Eurasia, b) North America, c)

South America and d) Africa for CloudSat, MERRA, MERRA-2, ERA-Interim and JRA-55 over

the time period 2007-2016. Mountain snow is in blue and non-mountain snow is in yellow.

