# Peer review of "How much snow falls in the world's mountains?"

_The Cryosphere, 2019_

## Referee Comment (RC1) · Anonymous Referee #1 · 29 Feb 2020

This paper discusses several assessments of snowfall accumulation in mountainous areas over the globe. Apart from one observational dataset (CloudSat), also several reanalyses datasets are considered. The paper is short and limited to giving an estimate of mountainous snowfall within the different datasets. Some short explanations for specific behavior are given. The methodology in the paper is rigor, but the results / conclusions are not very exciting or novel. I also question the relevance of the main conclusions of the paper.

My main comments:

- Mountainous precipitation or snowfall is very difficult to capture in models or reanaly-

sis. The precipitation scheme, orography, horizontal and vertical resolution and large-scale forcing all highly influence how precipitation develops and where precipitation is falling. The range of snowfall in the reanalyses is also extremely high (from 489 till 1891 mm (table 1)) which makes it almost impossible to state anything about 'how much snowfall is falling in the world's mountains' based on these datasets.

- The second research question the author's want to answer is 'what percentage of continental snow falls on mountainous regions?'. While reading the paper, I was wondering why this number matters? In the conclusions, the authors state that it is important for researchers who use snowfall estimates from reanalyses or observations, but I don't see how the 4-5% can help these researchers... The only conclusion I can draw from this analysis is that this percentage is similar in reanalysis and CloudSat, which states that the large-scale precipitation events are well captured in reanalyses, which is expected since these processes are assimilated herein. Is there another extra value of this result?

- Generally, I think the authors maybe have to rethink the scope of the paper. In my opinion, there are two options which are both already a bit discussed in the outlook of this paper. Option 1 would be to focus on CloudSat only, making advantage of the high-resolution product it offers and discuss in more detail mountainous or orographical snowfall by zooming in on specific features or large-scale processes and see how well these are captured by CloudSat. Option 2 would be to include other models (e.g. CMIP5?) and focus on the differences between these models in mountainous regions. This would also have a much higher impact and relevance for the scientific community.

Smaller comments:

- L76: This section deals about previous snowfall research using reanalyses. This section is very short compared to CloudSat. Has there been no more research on snowfall / precipitation in mountainous areas using reanalyses which could be added here?

- L166: The study of Maahn et al. (2014) is used to state that the height acquisition level of CloudSat has no huge influence on the ground precipitation estimate. However, this study focusses on polar regions. I think conditions might be very different for other mountainous regions. It is difficult to prove this of course for other regions without ground-based observations. Maybe add a line which refers to Grazioli et al. (2017) which gives a vertical profile of ERA-Interim precipitation compared to observations and clearly shows differences between both.

- Are any of the CloudSat observations assimilated in any of the reanalyses used? If this is the case, it should be noted and the results should be discussed with this in mind

- The discussion in section 4 is in my opinion a bit too short. Some results are discussed, but only a few times the behavior is explained and put in relation with the results of previous studies. For example, on line 275, it is stated that JRA-55 underestimates the intensity of snowfall. But what is the cause of this? Is JRA-55 not holding enough moisture? Is only snowfall underestimated? Are the number of events similar, but is there just too little precipitation? These are interesting features that are currently missing in the discussion, which might be retrieved from literature or small extra analyses. The same is true for line 285

- Figure 2/3: why is mm/month shown? Would mm/year not be more appropriate?

- Figure 4: the frequency of occurrence could maybe be replaced by events/year? This is more easy to understand by readers

---

## Author Comment (AC1) · 3 Mar 2020

Dear anonymous reviewer,

Thank you very much for the time you took to read and review our article. I will answer your comments with more details later but I would like to answer some of them now, specifically the ones about the novelty and the scope of the paper. Although I agree with some parts of your critics, I also strongly disagree with some of them and would like to explain why.

Based on my experience presenting these results, I think the scope of the paper should

not change but could certainly be better explained. We had very limited funding to work on this article and we tried to do a paper that would both be useful for the people developing the different datasets and the ones using them. Some of the users are part of the climate community and might be aware of the differences and biases in these datasets them but some of them are not (e.g. impact modelers) and they are most of the time very surprised by these differences. In an impact model, these differences can make a huge difference. Even in the climate community, researchers who know these datasets were usually very surprised by our results (e.g. MERRA2 South America). I have presented this work in a couple of conferences and it clearly appeared to us that these results were useful to a lot of people working with snow datasets from the climate community or other communities. So I agree with you, we know all these datasets but we need more papers giving a simple comparison of them to be aware of their limitations and abilities in different regions. Furthermore, I agree with you that more work could be done for understanding the differences between the differences between the datasets but I also think we already gave some interesting insights to the people developing the datasets. And again, with the limited time and funding we had, we could not go much further.

About the novelty, I think this paper is novel in different ways: 1) because all these reanalyses and CloudSat data had not been compared before in terms of snowfall on all these regions at the same time, 2) because they had not been compared before in terms of mountainous and non-mountainous snowfall.

So in general, I think we could clarify the aim of the paper in the abstract and the introduction as well as improving the section discussing the results, including some of the limitations you mentioned for example (height acquisition Maahn et al. 2014), however, I really think we should keep the scope of the paper as it is one because it is very useful for a number of researchers.

I am of course open to more discussion about this.

Anne Sophie.

---

## Referee Comment (RC2) · Jean-Baptiste Madeleine (Referee) · 18 Mar 2020

This paper quantifies the fraction of total snowfall that falls in the world mountains as well as the absolute amount of snowfall in the mountains, based on the CloudSat radar and different reanalyses. It analyzes the different datasets and gives possible explanations for the differences seen in the datasets, especially as it comes to the absolute amount of snowfall. A significant effort was made to compare the different datasets on the same grid, rigorously. The paper is well written and informative, and I think it deserves to be published.

However, some points need to be analyzed in greater depth. I have one major comment

and many smaller changes I would like to see in the final version of the paper. This won't require new work on the data though (I believe).

My major comment is the following : the maps (Figure 2) are great, but not analyzed at all, and it is a shame, because they DO contain a lot of information. The authors say "the geographical distribution of mountain snowfall is similar between CloudSat and all the reanalyses", but I disagree. There are many interesting differences. I think the authors must work more on the maps, by considering for example maps of the differences between the different datasets, or by computing mean RMS errors between each reanalysis and CloudSat (even though I understand CloudSat has its own uncertainties). For example, in the case of MERRA-2 (which clearly stands out), there is a lot of snowfall over the mountains of eastern Russia and Kamtchatka, more than for MERRA-1. Why ? JRA55 seems to miss a lot of the patterns too. Please elaborate more on these interesting maps !

Otherwise, here are some more minor comments :

l.34 : "the fraction of mountain snowfall" is ambiguous; the authors might want to change it to something like "the proportion of snow that falls in the mountains compared to the continent as a whole".

l.37 : I agree with the authors point regarding the large-scale forcings, and it is an interesting conclusion of the paper; all the models predict precipitation when air masses are converging. but I disagree on the point that the differences in the snowfall amounts result from differences "at smaller scales". As said line 327 in the conclusion, it is more likely due to differences in the physical parameterizations of the models, as well as subgrid-scale parameterizations of orographical effects.

l.84 : what do the authors mean by "is more realistic" ? and what does it have to do with the previous sentence ?

l.93 l.97 l.117 l.120 : you might be interested in the papers of my colleague, F. Lemon-

nier, on that subject :

CloudSat-inferred vertical structure of snowfall over the Antarctic continent F. Lemonnier, J.-B. Madeleine, C. Claud, C. Palerme, C. Genthon, T. L'Ecuyer, N. Wood JGR Atmospheres, doi:10.1029/2019JD031399, December 2019

Evaluation of CloudSat snowfall rate profiles by a comparison with in-situ micro rain radars observations in East Antarctica F. Lemonnier, J.-B. Madeleine, C. Claud, C. Genthon, C. Durán-Alarcón, C. Palerme, A. Berne, N. Souverijns, N. van Lipzig, I. V. Gorodetskaya, T. L'Ecuyer, N. Wood The Cryosphere Discuss., doi: 10.5194/tc-2018-236, March 2019

l.154 : I believe the Snow Retrieval Status (SRS) in release 5 was improved, and this might help select the profiles the authors use, especially in mountainous regions where the ground clutter might affect the retrievals. I am not saying that the authors should use release 5 and redo everything from scratch (please don't !), but that they might want to check if release 5 gives different results or not, just in case !

l.167 : "somewhat compensated by the competing effects of evaporation and undetected shallow snowfall" ; I have not read Maahn et al. (2014), but this sounds quite speculative to me. A lot can happen between the 1200m level and the surface, especially in mountains (slope winds, complex boundary layer). I think the authors should remain cautious about this point, and not say there is some kind of compensation of errors.

l.168 : this should be said earlier, when describing the CloudSat dataset.

l.174 : "less than about 15% at the surface"; what is "the surface" here ? the 1200m level ?

l.189 : "assimilates" > uses, is based on

l.199 : "while CloudSat started in 2007" this should be said earlier, when describing the CloudSat dataset.

l.206 : "based on the Kapos et al. (2000) definition" : could the authors summarize the criteria that define a mountainous terrain ?

l.233 : "In spite of these differences, the geographical distribution of mountain snowfall is similar between CloudSat and all the reanalyses" : as mentioned above in my major comment, I disagree, we see large differences between the different datasets, and these spatial differences might be part of the reason why the absolute amount of snowfall differs between them.

l.258 to 261 : does this mean that the CloudSat estimate, which is already high, is probably a lower bound, because it might miss some large events ? if so, this should be said in the text.

l.268 : "To ease the comparison between the different datasets" I don't understand why the amounts are normalized; to me it makes things more difficult to understand, with very different y axes. Are the authors sure it is the best way to represent this ?

Table 1 : I don't understand the row entitled "Global" : for example, 1763/43403 means that when the four continents are put together, 1763 cubic km per year of snow falls in the mountains (i.e. the sum of the rates for the four continents, which is not always exactly the case by the way...), but I don't understand the number "43403"; does it include Greenland and Antarctica ? it is much bigger than the sum of all the snowfall amounts. Please clarify.

Figure 4 : How is this frequency computed exactly and how comes this is so different between the different continents ? Please clarify.

Typos

l.57 : "the response of" can be removed

l.288 : "for MERRA-2", remove "for"

l.312 : "for researchers for"

[Figure]

l.317 : that THEY have difficulties ?

Figure 4, y axis, upper left panel : occurence > occurrence

---

## Short Comment (SC1) · 19 Mar 2020

Dear Jean-Baptiste,

Thank you very much for your positive and constructive comments. They are very helpful. I will answer the different comments you have at this end of the process but I wanted to answer some of them now.

1) About the maps that are not analyzed enough. Thank you very much it is a very good point especially since the idea of this article started by seeing some very large differences in the different datasets in some specific regions (e.g. South America) and

wondering if the differences were "global". So, yes we will definitively comment more some cases or regions and add more analyses in this section of the article.

2) About the confusion with the "fraction of mountain snowfall" and the suggestion to use "the proportion of snow that falls in the mountains compared to the continent as a whole", that's a very good point. It was difficult to name the different variables, it changed a couple of times in the different versions of the paper and we tried to come up with a short name but if it is not clear for the reader it does not work. So I will work on this in the new version of the paper.

3) About the hypothesis on large-scale versus small-scale, thank you very much. And I think we were at least partly thinking about the differences induced but the different parameterizations to explain the differences in small-scales but we need to be more clear about this hypothesis in the new version of the paper.

Anne Sophie.

————————————————————

---

## Author Comment (AC2) · 15 Apr 2020

Detailed Answer to reviewer #2: A version in pdf of these comments has been added in supplement This paper quantifies the fraction of total snowfall that falls in the world mountains as well as the absolute amount of snowfall in the mountains, based on the CloudSat radar and different reanalyses. It analyzes the different datasets and gives possible explanations for the differences seen in the datasets, especially as it comes to the absolute amount of snowfall. A significant effort was made to compare the different datasets on the same grid, rigorously. The paper is well written and informative, and I think it deserves to be published.

[Figure]

However, some points need to be analyzed in greater depth. I have one major comment and many smaller changes I would like to see in the final version of the paper. This won't require new work on the data though (I believe). Thank you very much for this constructive review, the major and minor comments you mentioned will be included in the new version of the paper if it is accepted. Your comments will be answered in blue in the following text.

Major comments: My major comment is the following: the maps (Figure 2) are great, but not analyzed at all, and it is a shame, because they DO contain a lot of information. The authors say "the geographical distribution of mountain snowfall is similar between CloudSat and all the reanalyses", but I disagree. There are many interesting differences. I think the authors must work more on the maps, by considering for example maps of the differences between the different datasets, or by computing mean RMS errors between each reanalysis and CloudSat (even though I understand CloudSat has its own uncertainties). For example, in the case of MERRA-2 (which clearly stands out), there is a lot of snowfall over the mountains of eastern Russia and Kamtchatka, more than for MERRA-1. Why ? JRA55 seems to miss a lot of the patterns too. Please elaborate more on these interesting maps ! We agree with you, Figure 2 needs and deserves more explanation and we agree that there are many interesting differences. In the potential next version of the article, we can add more discussion and go deeper in the analysis. We have done it for South America as mentioned in Sections 4 and 5 but we could talk more about some regions that stand out such as eastern Russia. A plot of the differences between CloudSat and the different reanalyses can be added to the paper to help us in the analysis.

Minor comments: l.34 : "the fraction of mountain snowfall" is ambiguous; the authors might want to change it to something like "the proportion of snow that falls in the mountains compared to the continent as a whole". Yes, thank you. This will modified in the next version of the article. l.37 : I agree with the authors point regarding the large-scale forcings, and it is an interesting conclusion of the paper; all the models predict precipi-
tation when air masses are converging. but I disagree on the point that the differences in the snowfall amounts result from differences "at smaller scales". As said line 327 in the conclusion, it is more likely due to differences in the physical parameterizations of the models, as well as subgrid-scale parameterizations of orographical effects. We totally agree with you and that's also what we thought when we saw the results but apparently our formulation of this idea was not correct. We will revise this part of the text to be more explicit.

l.84 : what do the authors mean by "is more realistic" ? and what does it have to do with the previous sentence ? This sentence has been reformulated, so it follows more logically the previous one.

l.93 l.97 l.117 l.120 : you might be interested in the papers of my colleague, F. Lemonnier, on that subject : CloudSat-inferred vertical structure of snowfall over the Antarctic continent F. Lemonnier, J.-B. Madeleine, C. Claud, C. Palerme, C. Genthon, T. L'Ecuyer, N. Wood JGR Atmospheres, doi:10.1029/2019JD031399, December 2019 Evaluation of CloudSat snowfall rate profiles by a comparison with in-situ micro rain radars observations in East Antarctica F. Lemonnier, J.-B. Madeleine, C. Claud, C. Genthon, C. Durán-Alarcón, C. Palerme, A. Berne, N. Souverijns, N. van Lipzig, I. V. Gorodetskaya, T. L'Ecuyer, N. Wood The Cryosphere Discuss., doi: 10.5194/tc-2018- 236, March 2019 Thank you, these references will be added to the text in the related section.

l.154 : I believe the Snow Retrieval Status (SRS) in release 5 was improved, and this might help select the profiles the authors use, especially in mountainous regions where the ground clutter might affect the retrievals. I am not saying that the authors should use release 5 and redo everything from scratch (please don't !), but that they might want to check if release 5 gives different results or not, just in case ! Yes, we can include a couple of sentences on how the last two versions of CloudSat differ.

l.167 : "somewhat compensated by the competing effects of evaporation and undetected shallow snowfall" ; I have not read Maahn et al. (2014), but this sounds quite

speculative to me. A lot can happen between the 1200m level and the surface, especially in mountains (slope winds, complex boundary layer). I think the authors should remain cautious about this point, and not say there is some kind of compensation of errors. This sentence will reformulated to be more conditional.

l.168 : this should be said earlier, when describing the CloudSat dataset. Yes, it is now said at the beginning of this section.

l.174 : "less than about 15% at the surface"; what is "the surface" here ? the 1200m level ? This will be clarified in the next version of the article.

l.189 : "assimilates" > uses, is based on Thank you, this sentence will be reformulated.

l.199 : "while CloudSat started in 2007" this should be said earlier, when describing the CloudSat dataset. Yes, this will be said earlier in the text, in the section concerning CloudSat. l.206 : "based on the Kapos et al. (2000) definition" : could the authors summarize the criteria that define a mountainous terrain ? We have tried to summarize their technic in the text adding more explanations. Their technic was quite complicated so it was difficult to add a general description, but more explanation will be added on the technic Kapos et al. (2000) used to define a mountainous terrain.

l.233 : "In spite of these differences, the geographical distribution of mountain snowfall is similar between CloudSat and all the reanalyses" : as mentioned above in my major comment, I disagree, we see large differences between the different datasets, and these spatial differences might be part of the reason why the absolute amount of snowfall differs between them. Yes, this point will be clarified and more analysis will be added to the text for Figure 2.

l.258 to 261 : does this mean that the CloudSat estimate, which is already high, is probably a lower bound, because it might miss some large events ? if so, this should be said in the text. CloudSat is certainly missing some large events but we hope that, as we have done this work on many years, the sample size is big enough to avoid a

big effect on CloudSat estimates.

l.268 : "To ease the comparison between the different datasets" I don't understand why the amounts are normalized; to me it makes things more difficult to understand, with very different y axes. Are the authors sure it is the best way to represent this ?

We first tried to look at the results without the normalization and we could not compare the results between the different datasets. The snowfall estimates are so different between the different datasets that we needed to include the normalization by grid points. For some datasets, it snows on much more grid points than other ones, but it does not snow a lot for example. We also wanted to normalize it by the grid cells to have a better feeling of what is happening in each reanalysis as they are based on models.

Table 1 : I don't understand the row entitled "Global" : for example, 1763/43403 means that when the four continents are put together, 1763 cubic km per year of snow falls in the mountains (i.e. the sum of the rates for the four continents, which is not always exactly the case by the way...), but I don't understand the number "43403"; does it include Greenland and Antarctica ? it is much bigger than the sum of all the snowfall amounts. Please clarify. Yes Global includes Greenland and Antarctica but this will be clarified in the text and in the Table.

Figure 4 : How is this frequency computed exactly and how comes this is so different between the different continents ? Please clarify. Yes, this can be clarified in the potential new version of the article.

Typos: l.57 : "the response of" can be removed Thank you, it will be removed. l.288 : "for MERRA-2", remove "for" Thank you, it will be corrected. l.312 : "for researchers for" Thank you, it will be corrected now. l.317 : that THEY have difficulties ? Thank you, it will be corrected. Figure 4, y axis, upper left panel : occurence > occurrence Thank you, this has been corrected in the text.

[Figure]

Please also note the supplement to this comment:
https://www.the-cryosphere-discuss.net/tc-2019-302/tc-2019-302-AC2-supplement.pdf

---

## Author Comment (AC3) · 15 Apr 2020

Dear Editor,

it is the first time I submit an article in Cryosphere, I just want to make sure I have done the right things in the last step.

During the interactive discussion with reviewers, I had answered the comments from the two reviewers that added reviews for our article. It was then indicated that I only needed to add a more detailed answer for reviewer #2. I did that today in a pdf file that I added as a supplement. My answers are also included as a plain text but the format

does not allow reading it easily. I join you here the pdf here, please let me know if I misunderstood something.

Best,

Anne Sophie.

Please also note the supplement to this comment:
https://www.the-cryosphere-discuss.net/tc-2019-302/tc-2019-302-AC3-supplement.pdf

---

## Author Response (AR2)

**Answer to reviewers for article:**

**"How much snow falls in the world's mountains? A first look at mountain snowfall estimates in A-train observations and reanalyses"**

For the reviewers and the editor: Line and page numbers are indicated for each comment based on the version of the manuscript with markups.

**Reviewer #1:**

This paper discusses several assessments of snowfall accumulation in mountainous areas over the globe. Apart from one observational dataset (CloudSat), also several reanalyses datasets are considered. The paper is short and limited to giving an estimate of mountainous snowfall within the different datasets. Some short explanations for specific behavior are given. The methodology in the paper is rigor, but the results / conclusions are not very exciting or novel. I also question the relevance of the main conclusions of the paper.

**General comments:**

- Mountainous precipitation or snowfall is very difficult to capture in models or reanalysis. The precipitation scheme, orography, horizontal and vertical resolution and largescale forcing all highly influence how precipitation develops and where precipitation is falling. The range of snowfall in the reanalyses is also extremely high (from 489 till 1891 mm (table 1)) which makes it almost impossible to state anything about 'how much snowfall is falling in the world's mountains' based on these datasets.
- The second research question the author's want to answer is 'what percentage of continental snow falls on mountainous regions?'. While reading the paper, I was wondering why this number matters? In the conclusions, the authors state that it is important for researchers who use snowfall estimates from reanalyses or observations, but I don't see how the 4-5% can help these researchers: : : The only conclusion I can draw from this analysis is that this percentage is similar in reanalysis and CloudSat, which states that the large-scale precipitation events are well captured in reanalyses, which is expected since these processes are assimilated herein. Is there another extra value of this result?
- Generally, I think the authors maybe have to rethink the scope of the paper. In my opinion, there are two options which are both already a bit discussed in the outlook of this paper. Option 1 would be to focus on CloudSat only, making advantage of the high-resolution product it offers and discuss in more detail mountainous or orographical snowfall by zooming in on specific features or large-scale processes and see how well these are captured by CloudSat. Option 2 would be to include other models (e.g. CMIP5?) and focus on the differences between these models in mountainous regions. This would also have a much higher impact and relevance for the scientific community.

Although I agree with some parts of your critics (more discussion about the differences between the datasets, cf. new Figure 3), I also strongly disagree with some of them and would like to explain why. Based on my experience presenting these results, I think the scope of the paper should not change but could certainly be better explained.

We think this paper is useful for number of communities: for example, for people developing the different datasets and for researchers using them. Some of the users are part of the climate community and might be aware of the differences and biases in these datasets but some of them are not (e.g. impact modelers) and they are most of the time very surprised by and interested in these differences. In an impact model, these differences can make a huge difference. Even in the climate community, researchers who know these datasets are usually very surprised by our results (e.g. MERRA2 South America). I have presented this work in a couple of conferences and it clearly appeared to us that these results were useful to a lot of people working with snow datasets from the climate community or other communities. So, I agree with you, we know all these datasets, but we need more papers giving a simple comparison of them to be aware of their limitations and abilities in different regions. A better knowledge on how these measurements compare is also very important because mountain snow is critical for water runoff, which is used for example as fresh water or for agriculture but is also susceptible to create flooding.

About the novelty, I think this paper is novel in several different ways: 1) because all these reanalyses and CloudSat data had not been compared before in terms of snowfall on all these regions at the same time, 2) because they had not been compared before in terms of mountainous and non-mountainous snowfall, 3) there have never been an observational estimate of the amount of snow that falls in the major mountain ranges and its distribution throughout them. Thus, this study provides a critical benchmark of the current climate and the accuracy to which we know it against which future trends can be monitored and predictions can be assessed.

So in general, I think we could clarify the aim of the paper in the abstract and the introduction as well as improving the section discussing the results, including some of the limitations you mentioned for example (height acquisition Maahn et al. 2014), however, I really think we should keep the scope of the paper as it is one because it is very useful for a number of researchers.

**Smaller comments:**

- L76: This section deals about previous snowfall research using reanalyses. This section is very short compared to CloudSat. Has there been no more research on snowfall / precipitation in mountainous areas using reanalyses which could be added here?

Yes we agree, this is certainly something lacking in the article. More results from recent publications have been added to the Introduction (p. 4-5, l.90-105):

1) Wang, C., Graham, R. M., Wang, K., Gerland, S., and Granskog, M. A.: Comparison of ERA5 and ERA-Interim near-surface air temperature, snowfall and precipitation over Arctic sea ice: effects on sea ice thermodynamics and evolution, The Cryosphere, 13, 1661–1679, https://doi.org/10.5194/tc-13-1661-2019, 2019.

2) Orsolini, Y., Wegmann, M., Dutra, E., Liu, B., Balsamo, G., Yang, K., de Rosnay, P., Zhu, C., Wang, W., Senan, R., and Arduini, G.: Evaluation of snow depth and snow cover over the Tibetan Plateau in global reanalyses using in situ and satellite remote sensing observations, The Cryosphere, 13, 2221–2239, https://doi.org/10.5194/tc-13-2221-2019, 2019.

3) Cohen L. and S. Dean, 2013: Snow on the Ross Ice Shelf: comparison of reanalyses and observations from automatic weather stations. The Cryosphere, 7, 1399-1410. Doi:10.5194/tc-7-1399-2013.

4) Liu Y. and S.A. Magulis, 2019: Deriving bias and uncertainty in MERRA-2 snowfall precipitation over High Mountain Asia. Front. Earth. Sci., https://doi.org/10.3389/feart.2019.00280

- L166: The study of Maahn et al. (2014) is used to state that the height acquisition level of CloudSat has no huge influence on the ground precipitation estimate. However, this study focusses on polar regions. I think conditions might be very different for other mountainous regions. It is difficult to prove this of course for other regions without ground-based observations. Maybe add a line which refers to Grazioli et al. (2017) which gives a vertical profile of ERA-Interim precipitation compared to observations and clearly shows differences between both.
Yes thank you, the results from Grazioli et al. (2017) has been added to the text to temper the results from Maahn et al. (2014), (p.9, l.201-204).

- Are any of the CloudSat observations assimilated in any of the reanalyses used? If this is the case, it should be noted and the results should be discussed with this in mind
Yes, I agree with you, this is a very important point. CloudSat is not assimilated into any of the four reanalyses and this is now clearly mentioned in the text, (p.11, l. 245-246).

- The discussion in section 4 is in my opinion a bit too short. Some results are discussed, but only a few times the behavior is explained and put in relation with the results of previous studies. For example, on line 275, it is stated that JRA-55 underestimates the intensity of snowfall. But what is the cause of this? Is JRA-55 not holding enough moisture? Is only snowfall underestimated? Are the number of events similar, but is there just too little precipitation? These are interesting features that are currently missing in the discussion, which might be retrieved from literature or small extra analyses. The same is true for line 285
Yes, thank you, the discussion has now been slightly extended to include more references to previous work comparing the reanalysis datasets and showing some of their differences. For example, our previous work on the representation of clouds in the in different reanalyses showed some large differences between them and major biases for JRA-55 and MERRA-2, (p.18, l.396-407).

- Figure 2/3: why is mm/month shown? Would mm/year not be more appropriate?
In this type of study it seems that researchers are using different units, mm/month is one of them. We have also seen mm/day and mm/year. However, as our datasets are aggregated by month, mm/month seemed like a better choice for this study.
- Figure 4: the frequency of occurrence could maybe be replaced by events/year? This is more easy to understand by readers
Frequency of occurrence is a standard metric used in the remote sensing community, so we think we should keep this metric intact.

**Reviewer #2:**

This paper quantifies the fraction of total snowfall that falls in the world mountains as well as the absolute amount of snowfall in the mountains, based on the CloudSat radar and different reanalyses. It analyzes the different datasets and gives possible explanations for the differences seen in the datasets, especially as it comes to the absolute amount of snowfall. A significant effort was made to compare the different datasets on the same grid, rigorously. The paper is well written and informative, and I think it deserves to be published.

However, some points need to be analyzed in greater depth. I have one major comment and many smaller changes I would like to see in the final version of the paper. This won't require new work on the data though (I believe).

Thank you very much for the constructive and very helpful comments in your review. Most of them have been included in the text and have participated to increase the quality of the article.

**Major comments:**

My major comment is the following: the maps (Figure 2) are great, but not analyzed at all, and it is a shame, because they DO contain a lot of information. The authors say "the geographical distribution of mountain snowfall is similar between CloudSat and all the reanalyses", but I disagree. There are many interesting differences. I think the authors must work more on the maps, by considering for example maps of the differences between the different datasets, or by computing mean RMS errors between each reanalysis and CloudSat (even though I understand CloudSat has its own uncertainties).
For example, in the case of MERRA-2 (which clearly stands out), there is a lot of snowfall over the mountains of eastern Russia and Kamtchatka, more than for MERRA-1.
Why ? JRA55 seems to miss a lot of the patterns too. Please elaborate more on these interesting maps !

[Figure]

*__Figure 3:__ Spatial maps of the global cumulative mountain snowfall (mm/month/gridbox) over the High-mountains Asia for a) CloudSat, b) CloudSat minus MERRA, c) CloudSat minus MERRA-2, d) CloudSat minus ERA-Interim and d) CloudSat minus JRA-55, over the time period 2007-2016*

Yes, this is a very good point. In general, section 3.1 has been reorganized and is longer now. Figure 2 is now analyzed in more details and a new figure (Figure 3) has been added to this section to complement the analysis. Figure 3 shows the differences between CloudSat and the other datasets over the High-mountain Asia. To have a more interesting analysis of the differences between the datasets, we decided to focus on one region, (p.13-14, l.297-313).

**Minor comments:**

l.34 : "the fraction of mountain snowfall" is ambiguous; the authors might want to change it to something like "the proportion of snow that falls in the mountains compared to the continent as a whole".
Yes, thank you. This has been corrected in the text where it is mentioned, (p.2, l.34-35)

l.37 : I agree with the authors point regarding the large-scale forcings, and it is an interesting conclusion of the paper; all the models predict precipitation when air masses are converging. but I disagree on the point that the differences in the snowfall amounts result from differences "at smaller scales". As said line 327 in the conclusion, it is more likely due to differences in the physical parameterizations of the models, as well as subgrid-scale parameterizations of orographical effects.
We agree with you on the point but apparently, we did not explain it correctly. This hypothesis has been reformulated in the article and is hopefully clearer and in phase with your explanation, (p.2, l.39-41; p.20, l.441-442).

l.84 : what do the authors mean by "is more realistic" ? and what does it have to do with the previous sentence ?
This sentence has been reformulated, so it follows more logically the previous one, (p.4, l.90-93).

l.93 l.97 l.117 l.120 : you might be interested in the papers of my colleague, F. Lemon-nier, on that subject :
CloudSat-inferred vertical structure of snowfall over the Antarctic continent:
F. Lemonnier, J.-B. Madeleine, C. Claud, C. Palerme, C. Genthon, T. L'Ecuyer, N. Wood JGR Atmospheres, doi:10.1029/2019JD031399, December 2019
Evaluation of CloudSat snowfall rate profiles by a comparison with in-situ micro rain radars observations in East Antarctica
F. Lemonnier, J.-B. Madeleine, C. Claud, C. Genthon, C. Durán-Alarcón, C. Palerme, A. Berne, N. Souverijns, N. van Lipzig, I. V. Gorodetskaya, T. L'Ecuyer, N. Wood The Cryosphere Discuss., doi: 10.5194/tc-2018-236, March 2019
Thank you, these references have been added to the text in the related section (p.5, l. 114).

l.154 : I believe the Snow Retrieval Status (SRS) in release 5 was improved, and this might help select the profiles the authors use, especially in mountainous regions where the ground clutter might affect the retrievals. I am not saying that the authors should use release 5 and redo everything from scratch (please don't !), but that they might want to check if release 5 gives different results or not, just in case !

You might be referring to the ground clutter contamination affecting the snowfall retrieval. One part of the snow retrieval status variable can help diagnose when ground clutter could cause the snowfall retrieval results to be in error. The snow retrieval status variable is evaluated in the same way in both R04 and R05. One of the differences between R04 and R05, however, is that the digital terrain elevation map used in R05 is improved compared to the map in R04. As a result, the retrieval does a better job of identifying the position of the lowest clutter-free bin and there are fewer retrievals contaminated with ground clutter. Globally, the long-term annual snowfall amount changes from 76.0 mm/y in R04 to 75.7 mm/y in R05, but there are locations that are more strongly impacted. The areas that are most strongly affected are Greenland, the edges of the Tibetan plateau, and some mountainous points in Antarctica. From the mountain mask figure in the paper, Greenland and Antarctica have no locations that are classified as mountainous. The Tibetan plateau is part of a much larger area of Asia that is classified as mountainous. Because of this, we believe the impacts of R05 on the results of the paper would be negligible. This has been clarified in the text, (p.9, l.189-193).

l.167 : "somewhat compensated by the competing effects of evaporation and undetected shallow snowfall"; I have not read Maahn et al. (2014), but this sounds quite speculative to me. A lot can happen between the 1200m level and the surface, especially in mountains (slope winds, complex boundary layer). I think the authors should remain cautious about this point, and not say there is some kind of compensation of errors.
This sentence has been reformulated to be more conditional (p.10, l.209).

l.168 : this should be said earlier, when describing the CloudSat dataset.
Yes, it is now said at the beginning of this section, (p.8, l.174).

l.174 : "less than about 15% at the surface"; what is "the surface" here ? the 1200m level ?
The surface is considered as 1.2 km and this is now clarified in the text, (p.10, l.218). Instantaneous CloudSat quantitative precip retrievals (units of mm/h) are derived from the first usable bin above the surface. However, ECMWF temperature profiles are used to determine the probable *surface* phase (rain versus mixed versus snow) to account for possible melting in the radar "blind zone". So surface precipitation phase refers to ground-level. But the precipitation rate at ground level is not corrected in any way - it is the precipitation rate derived from the first usable bin above the surface (~1.2 km above ground level).

l.189 : "assimilates" > uses, is based on
Thank you, this sentence has been reformulated, p.11 (l.236).

l.199 : "while CloudSat started in 2007" this should be said earlier, when describing the CloudSat dataset.
Yes, this is now said earlier in the text, in the section concerning CloudSat, (p.9, l.185).

l.206 : "based on the Kapos et al. (2000) definition" : could the authors summarize the criteria that define a mountainous terrain ?
We have tried to summarize their technic in the text adding more explanation but their technique was quite complicated so it was difficult to add a general description (p.12, l.257-260).

l.233 : "In spite of these differences, the geographical distribution of mountain snowfall is similar between CloudSat and all the reanalyses" : as mentioned above in my major comment, I disagree, we see large differences between the different datasets, and these spatial differences might be part of the reason why the absolute amount of snowfall differs between them.

Yes, this point is now clarified, and more analysis has been added to the text for Figure 2, and a Figure 3 commenting some of the differences between the datasets has been added, (p.13-14, l. 297-313).

l.258 to 261: does this mean that the CloudSat estimate, which is already high, is probably a lower bound, because it might miss some large events? if so, this should be said in the text. CloudSat may be missing a few large events but as this analysis over many months and years, the effect of these few events should still be limited. This has now been clarified in the text, (p.16, l.349-350).

l.268: "To ease the comparison between the different datasets" I don't understand why the amounts are normalized; to me it makes things more difficult to understand, with very different y axes. Are the authors sure it is the best way to represent this?
We understand your point but without normalization the results were hard to interpret. The snowfall estimates are so different between the different datasets that we needed to include the normalization by grid points. For example, for some datasets, you can have the same amount of snow over an area, however, for one dataset it snows on much more grid points than other ones and it snows less than the other dataset. So, we needed a way to compare how the amount of snow was distributed over the area examined.

Table 1 : I don't understand the row entitled "Global" : for example, 1763/43403 means that when the four continents are put together, 1763 cubic km per year of snow falls in the mountains (i.e. the sum of the rates for the four continents, which is not always exactly the case by the way...), but I don't understand the number "43403"; does it include Greenland and Antarctica? it is much bigger than the sum of all the snowfall amounts. Please clarify.
Yes, Global includes Greenland and Antarctica. This was indicated in the text in the Section 2 but this is clearly not enough so now it is also indicated in the legend of Table 1, p.31, l.687-688).

Figure 4 : How is this frequency computed exactly and how comes this is so different between the different continents ? Please clarify.
For each grid box, we counted every instance of time (call this total events). we also created a separate variable to count up every instance of time where snow > 0 (call this snow events). At the end of iterating through every instance of time, for each grid box, we computed the ratio of snow events to total events (snow events / total events).
The difference between continents in frequency of occurrence of snowfall is due to numerous factors. These differences could be due to proximity to bodies of water, the spatial coverage of mountain ranges (x vs. y on a map; the Andes, for example, are much narrower in the 'y' direction), possibly height of mountain ranges, latitude probably plays a factor as well., prevailing winds, synoptic weather patterns, climate-scale oscillations, terrain gradient, etc. could also influence the results.

**Typos:**
l.57 : "the response of" can be removed
Thank you, it has been removed, (p.3, l.53).
l.288 : "for MERRA-2", remove "for"
Thank you, it is corrected, (p.17, l.376).
l.312 : "for researchers for"
Thank you, it is corrected now, (p.19., l.423)
l.317 : that THEY have difficulties ?
Thank you, it is corrected, (p.19, l.429).

Figure 4, y axis, upper left panel : occurence > occurrence
Thank you, this has been corrected in the text.

[revised manuscript text omitted]